# Dense Pixel-level Beef Cattle Instance Segmentation Using a Fully Convolutional Network

**Aram Ter-Sarkisov, Robert Ross & John Kelleher,**
Department of Computer Science, Dublin Institute of Technology, Dublin, Republic of Ireland
{alex.ter-sarkisov, robert.ross, john.d.kelleher}@dit.ie

**Bernadette Earley & Michael Keane**
Animal and Bioscience Research Department, Animal and Grassland Research and Innovation Centre, TEAGASC, Grange, Dunsany, Co.Meath, C15 PW93, Republic of Ireland
{Bernadette.Earley, Michael.Keane}@teagasc.ie

## Abstract

We present an instance segmentation algorithm trained and applied to a video of a group of heifers recorded in a winter finishing feedlot. We transform a fully convolutional class segmentation network into an instance segmentation network that learns to label each instance of a heifer separately. We introduce two new modules that teach the network to output a single prediction for every animal. These results are an early contribution towards behaviour analysis in winter finishing beef cattle for early detection of welfare-related problems, namely lameness.

## 1 Introduction

The background to this paper is an ongoing project focused on developing machine learning solutions to help in studies of animal welfare in an on-farm setting. In this particular instance our goal is to support studies of cattle welfare in a winter finishing feedlot. A specific challenge in these studies is to track the movement and behaviour of individual cattle in pens over a prolonged period. Adopting a computer vision approach to this challenge part of the solution involves segmenting individual cattle in each frame in a video. This motivates the development of an instance segmentation model. Class and instance segmentation of images are two different problems: the former (see FCN Long et al. (2015), CRF-RNN Zheng et al. (2015), DeepLab Noh et al. (2015); Chen et al. (2017; 2016) labels every instance of a class in an image with the same label, the latter (see MaskRCNN He et al. (2017), FCIS Li et al. (2016) and MNC Dai et al. (2016)) also identifies all the instances of a class in an image but crucially distinguishes between different instances of the class, labelling each identified instance with a distinct label. State-of-the-art instance segmenter, MaskRCNN uses region proposals (RPN) to find bounding box coordinates for separate objects and generates a class-specific mask within the bounding box. Although both MaskRCNN and FCN perform well on Pascal VOC 2012 and MS COCO 2017 (Everingham et al. (2010); Lin et al. (2014), they performed poorly when we applied them to the task we are interested, namely the segmentation of cattle in a winter finishing facility. This drop in performace was due to the frequent partial occlusion, similarity between instances and other challenges. Therefore we extracted a new labelled dataset from the video and finetuned both MaskRCNN and FCN on it. To convert FCN8s into an instance segmentation network, we introduce two new modules: MaskExtractor and BadOverlapsExtractor.

## 2 Dataset and Network Architecture

The raw video was recorded at a winter finishing feedlot over a two-week period with a camera installed at a fixed angle directed towards enclosures with 10 heifers in each, a total of 8 enclosures, see Keane et al. (2017). We selected one of the enclosures for the construction of the dataset. The dataset construction and the naive segmentation algorithm were presented in Ter-Sarkisov et al.

**Input:** Argmax binary output mask **B** size HxWx1
**Input:** Ground truth mask **C** size HxWx1
Get Intersect over Union (IoU) matrix for all predict blobs in **B** and ground truth blobs in **C**
**for** *each output blob* **b** $\in$ **B do**
    Rank IoUs of this prediction in the decreasing order
    *FoundBestOverlap*=**False**
    **for** *each ground truth blob* **c** $\in$ **C do**
        Get IoU(**b**, **c**)
        **if** *IoU (*$\mathbf{b}, \mathbf{c}$*) >0* **then**
            **if** *IoU (*$\mathbf{b}, \mathbf{c}$*)* $== \max IoU(:, \mathbf{c})$ **then**
                *FoundBestOverlap*=**True**
            **else**
                Set overlap pixels to ignore value (255)
        **else**
            **break**
    **end**
**end**
**Output:** Modified ground truth mask $\mathbf{C}^*$ with pixels in 'bad' overlaps set to ignore value

**Algorithm 1:** MaskExtractor

(2017). The resulting dataset was cropped and split into 4872 train and validation and 984 test images size 250x250 centered in every identified heifer.

FCN8s is transformed into an instance segmentation network by adding (either separately and together) two new modules described below. If there are more that two predictions for single animal or single prediction for two or more animals, non-maximum overlaps are declared 'bad'.

1. FCN8s + MaskExtractor (ME). Once the Intersect over Union (IoU) matrix is computed, we identify 'bad' overlaps between blobs and set pixels corresponding to them in the ground truth to ignore value. This changes the loss function and the derivatives during backpropagation calculation.

2. FCN8s + BadOverlapsExtractor (BOE). This module teaches the network to minimize output of 'bad' blobs. The *score* maps are connected with a 1x1 kernel to a single convolutional map. This map takes a pixelwise product with the argmax binary mask to obtain a sparse map that is fully connected to a 1x1 InnerProduct unit, i.e. one value, which is how the network learns to output the number of 'bad' blobs, called *output*. The ground truth is extracted by finding the number of overlaps between the argmax mask of *score* and ground truth mask. Both values are plugged in a Euclidean loss function.

Since ME is a tweak for the loss function and BOE is a subnetwork, they can be used together in the same network, FCN8s+ME+BOE. For comparison we finetune FCN8s alone and MaskRCNN (both full network and problem-specific layers only).

## 3 EXPERIMENTS AND RESULTS

All FCN-based models were trained using Caffe framework with ADAM optimizer (Kingma & Ba (2014)) with standard parameters and base learning rate of 0.00001 for 20K iterations on Tesla K40m GPU with 12 GB of VRAM. Since the networks are very large ($\sim$137M parameters), we used a small batch size of 4, and each iteration (feedforward+backpropagation) during training took 1.2 sec/iteration. As the testing procedure is identical to FCN (with fewer score maps, 2 instead of 21), the network required only 90 ms/image. MaskRCNN was finetuned from ResNet101-FPN weights pre-trained on MS COCO with the starting learning rate of 0.0002 and max/min image size 256x256 (as this is the closest to the 250x250 training images) for 20K iterations. All models were trained in one shot end-to-end.

All results before (for MaskRCNN and FCN8s) and after finetuning on our data (augmented with 64 images from Pascal VOC training set) are presented in Table 1. FCN8s with both ME and

```
Input: Argmax binary output mask B size HxWx1
Input: Ground truth mask C size HxWx1
NumBadOverlaps = 0
Get Intersect over Union (IoU) matrix for all predict blobs in B and ground truth blobs in C
for each ground truth blob c ∈ C do
    if max(IoU(:, c)) == 0 then
        NumBadOverlaps+=1
end
for each output blob b ∈ B do
    if max(IoU(b, :)) == 0 then
        NumBadOverlaps+=1
        skip
    else
        Rank IoUs of this prediction in the decreasing order
        FoundBestOverlap=False
        for each ground truth blob c ∈ C do
            Get IoU(b, c)
            if IoU (b, c) >0 then
                if IoU (b, c) == max IoU(:, c) then
                    FoundBestOverlap=True
                else
                    NumBadOverlaps+1
            else
                break
        end
end
Output: NumBadOverlaps
```

**Algorithm 2:** BadOverlapsExtractor

Table 1: Results on our test set, Pascal VOC and MS COCO validation set before and after finetuning to our data + Pascal VOC 2012 training set

| Model | Our test set | | Pascal VOC validation | | MS COCO validation | |
|---|---|---|---|---|---|---|
| | AP@0.5 | AP@0.5:0.95 | AP@0.5 | AP@0.5:0.95 | AP@0.5 | AP@0.5:0.95 |
| FCN8s | 0.041 | 0.010 | 0.520 | 0.293 | 0.441 | 0.235 |
| MaskRCNN | 0.365 | 0.053 | 0.764 | 0.621 | 0.563 | 0.417 |
| FCN8s-ft | 0.558 | 0.253 | 0.271 | 0.109 | 0.099 | 0.027 |
| MaskRCNN-heads-ft | 0.637 | 0.265 | 0.501 | 0.214 | 0.291 | 0.082 |
| MaskRCNN-ft | 0.687 | 0.277 | 0.298 | 0.120 | 0.255 | 0.089 |
| FCN8s+ME | 0.608 | 0.290 | 0.239 | 0.089 | 0.096 | 0.025 |
| FCN8s+BOE | 0.664 | 0.341 | 0.214 | 0.071 | 0.057 | 0.016 |
| FCN8s+ME+BOE | **0.694** | **0.352** | 0.234 | 0.068 | 0.057 | 0.017 |

BOE outperforms MaskRCNN (both fully trained and heads only) despite using a far shallower backbone network (FCN8s vs ResNet101). For reference, we also ran all finetuned networks on Pascal VOC and MS COCO validation sets, and it is obvious that performance deteriorated significantly. This is mostly due to a vast difference between our data, that has a persistent presence of a range of challenges, but lacks diversity (i.e. videos in the feedlot are recorded from a fixed angle 100% of the time).

## 4 CONCLUSIONS AND DISCUSSION

In this publicatin we presented a network that is built on top of FCN8s. We introduced two new network modules that convert a dense pixel-level class segmenter into an instance segmentation network with additional layers, that are only used in the training phase. We would like to extend these findings to the development of a real-time animal tracker using a recurrent neural network that would predict animal's behaviour and health condition.

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
