# OpenReview forum: "Dense Pixel-level Beef Cattle Instance Segmentation Using a Fully Convolutional Network"
_ICLR.cc/2018/Workshop — Reject_

### Official Review · AnonReviewer2 · 2018-03-06
**Missing important details, unconvincing results**

**Rating:** 3
**Confidence:** 5

**Review:**

* Overview*
The paper tackles instance segmentation for images of beef cattle.
Mostly the paper is badly written and important details of the proposed approach are missing. Overall the results are extremely underwhelming on the public benchmarks, and the gains on the in-house cattle dataset are very small. Most importantly, the novelty of the approach is non-existent. See below for clarifications.

* Details*
The authors build on FCN, a method designed for instance segmentation as they correctly mention in the intro. During the description of the approach, they mention that they predict blobs from the FCN output. There is no mention of what that is, how they obtain it from the semantic mask predictions. Unless I am missing this information, this is a crucial point. Moving beyond that important detail, the authors propose two modules, which are essentially hard negative mining (similar to Training Region-based Object Detectors with Online Hard Example Mining, Shrivastava et al., CVPR 2016) and another layer of classification.
If I were to ignore the extremely limited novelty of the proposed approach, the results (Table 1) are not compelling at all. The authors build on a shallow architecture and show underwhelming results on the public benchmarks. On top of that, the hyper-parameters used for computing published approaches are suboptimal (e.g. image size) which lead to lower performance than the ones reported by the original papers.

Last, while the above review sounds harsh I want to emphasize that it is quite legitimate to study a problem of particular interest (here image of beef cattle) and to care to obtain good results with low compute and at a low cost presumably. However, a publication is not necessarily justified based on the observations. I truly believe that there is people that will be interested in this approach (e.g. farmers, producers etc.) but a publication at an ICLR workshop is not the right venue for this paper.

---

> ### Public Comment · ~Alex_Ter-Sarkisov1 · 2018-03-21
> **Problem-specific network**
>
> The network was not designed to compete against leaders in MS COCO/Pascal datasets, it is specific to this problem, like tumor segmentation or similar. Results for MS COCO and Pascal were produced for reference, which is stated in the paper. On our test data finetuned state-of-the-art algorithm Mask R-CNN produces 68.7% AP at 0.5 threshold and 27.7% mAP (at 50%:95% thresholds) vs FCN+MaskExtractor+BadOverlapsExtractor producing 69.4% AP at 0.5 threshold and 35.2% mAP (at 50%:95% thresholds). Reported improvements in benchmark datasets are rarely larger.
>
> Giving all the details is simply intractable in a 3-page paper, it was intended to demonstrate the applicability and the potential of the idea of transforming (ignore pixels) and extracting additional information from the ground truth mask (count the number of 'bad' overlaps, e.g. 1 prediction/2+ cows or 2+ predictions/1 cow).  There are two additional modules, one that learns good predictions and one that learns which ones are 'bad'. They were intended to get the network to output single prediction (mask) per single observation (cow, in this case). This solution is perfectly transferable to any object vs background problem with heavy partial occlusion. I didn't use any negative hardmining: only positive blobs were segmented in the ground truth.

---

### Official Review · AnonReviewer1 · 2018-03-11
**Unclear presentation makes understanding of potential contributions difficult**

**Rating:** 2
**Confidence:** 5

**Review:**

The submission describes two possible modifications to an FCN-8s semantic segmentation network that supposedly enable the architecture to perform instance segmentation on a cattle surveillance data set.

In the opinion of this reviewer, little care went into preparation of this submission. Omission of crucial details makes understanding of what's going on very difficult. The problem is never defined clearly, and the description of the two modifications ('MaskExtractor' and 'BadOverlapsExtractor') is so condensed, and devoid of formulas or clarifying diagrams or pictures, that it is hard to follow. Both seem like fairly ad-hoc additions; especially then, proper justification is warranted.

It is never described where exactly the two modules are added to the existing pipeline. Both Algorithm 1 and 2 are never referenced in the text. It is unclear how Algorithm 2 fits to the description of the BadOverlapsExtractor module; I do not see any 1x1 kernels mentioned in Algorithm 2. It is unclear how backpropagation through these two modules might work.

Despite the lack of clarity, I believe that better results compared to a fine-tuned Mask R-CNN may have been achieved; the problem is clearly more constrained and quite different to what the pre-trained networks were trained on.
(On a side note, it seems the whole data set (training, validation & test sets) was created using data from one enclosure. Generalization to other enclosures may not work very well, due to different scene geometry and/or camera angles.)

Overall, I cannot recommend this submission for acceptance. I believe the clarity of presentation should be significantly improved.

There are several typos in this submission:
- "If there are more tha*n* two predictions...""
- "In this publicati*o*n we presented..."
- "MaskRCNN" should be "Mask R-CNN" (several times -- please cite names properly!)
- and potentially more

---

### Decision · Program_Chairs · 2018-03-20
**ICLR 2018 Workshop Acceptance Decision**

**Decision:**

Reject

**Comment:**

Based on the reviews, this paper has not been accepted for presentation at the ICLR workshop. However, the conversation and updates can continue to appear here on OpenReview.